# Identification of IL-18 and Soluble Cell Adhesion Molecules in the Gingival Crevicular Fluid as Novel Biomarkers of Psoriasis

**DOI:** 10.3390/life11101000

**Published:** 2021-09-23

**Authors:** Fernando Valenzuela, Javier Fernández, Constanza Jiménez, Daniela Cavagnola, Juan Felipe Mancilla, Jessica Astorga, Marcela Hernández, Alejandra Fernández

**Affiliations:** 1Department of Dermatology, Faculty of Medicine, Universidad de Chile, Santiago 8380453, Chile; 2Centro Internacional de Estudios Clínicos, Probity Medical Research, Santiago 8420383, Chile; javi.fernandez@uc.cl; 3Dermatology Unit, San José Hospital, Santiago 8380419, Chile; 4Department of Oral Pathology, Faculty of Dentistry, Universidad Andres Bello, Santiago 8370133, Chile; c.jimenezlizama@uandresbello.edu (C.J.); d.cavagnolacavagnola@uandresbello.edu (D.C.); j.mancillauribe@uandresbello.edu (J.F.M.); 5Laboratory of Periodontal Biology, Faculty of Dentistry, Universidad de Chile, Santiago 8380544, Chile; jv_astorga@med.uchile.cl (J.A.); mhernandezrios@odontologia.uchile.cl (M.H.); 6Department of Oral Pathology and Medicine, Faculty of Dentistry, Universidad de Chile, Santiago 8380544, Chile

**Keywords:** psoriasis, biomarkers, interleukin 18, cell adhesion molecules, gingival crevicular fluid, periodontal disease, skin diseases, E-selectin, intercellular adhesion molecule-1

## Abstract

Psoriasis is a chronic immunoinflammatory skin disease. Although its diagnosis is clinical, differences in the appearance and severity of lesions pose a challenge for clinicians worldwide. The use of accessible biomarkers for psoriasis could aid in the early diagnosis and treatment of the disease. To date, evidence on the analysis of gingival crevicular fluid (GCF) molecules as novel, accessible, and reliable biomarkers for psoriasis is limited. This cross-sectional study compared the GCF levels of IL-18, soluble (s)ICAM-1, and sE-selectin in psoriatic patients (n = 42) and healthy controls (n = 39). Individuals with psoriasis not undergoing treatment and healthy individuals were included independent of periodontal status. GCF samples were collected, and a multiplex bead immunoassay was performed to quantify the levels of the target molecules. Psoriatic patients presented higher concentrations of IL-18 and lower concentrations of sE-selectin compared to controls (*p* < 0.05). No differences were found in the levels of sICAM-1 between the two groups (*p* > 0.05). Psoriasis was associated with IL-18 and E-selectin levels regardless of periodontal status, age, and smoking habit (*p* < 0.05). The areas under the receiver operating characteristic curve (ROC) for IL-18 and sE-selectin were 0.77 and 0.68, respectively. In conclusion, IL-18 and sE-selectin levels in the GCF could be promising biomarker for psoriasis.

## 1. Introduction

Psoriasis is a chronic, non-communicable immunoinflammatory disease of the skin that significantly impacts patients’ quality of life. Its etiology remains unclear; nonetheless, research supports a decisive role of environmental factors and genetic predisposition in its development [1,2]. The worldwide prevalence of psoriasis ranges from 0.09% to 11.43%, with an average age of diagnosis between 47.7 and 58.7 years old and an incidence rate probably higher in women [3].

Major clinical signs of psoriasis include the formation of well-defined erythematous plaques and scabs on visible skin areas, as well as itching, stinging, and pain [1,4]. Currently, psoriasis is associated with an increased inflammatory burden, which facilitates the development of several immunoinflammatory conditions such as cardiovascular disease, metabolic syndrome, and periodontitis [1,5,6]. Therefore, psoriasis must be considered to be an immune-mediated systemic disorder and not a mere localized skin disease.

Dysregulation of the skin’s immune system is one of the main pathogenic mechanisms of psoriasis [7]. The activation of T helper 1 (Th1) and Th17 axes stimulates the overexpression of proinflammatory cytokines in the skin and serum [7,8,9]. Interleukin (IL) -1 and IL-18 are proinflammatory cytokines belonging to the IL-1 family, which is one of the key players in the development of psoriasis. IL-18 induces Th1 responses in the presence of IL-12 and the absence of Th2, and it also induces intercellular adhesion molecule-1 (ICAM-1) production [10,11]

In addition, the collective production of IL-18 and -23 stimulates the secretion of IL-17 by Th17 lymphocytes [9]. In this regard, higher IL-18 mRNA and protein levels have been reported in skin lesions and serum of psoriatic patients compared to healthy controls [12,13,14]. Furthermore, the serum levels of IL-18 in psoriatic patients have been positively correlated with the severity of the disease [13,14]. Altogether, these findings suggest that IL-18 could be a valuable biomarker for diagnosing and staging the severity of psoriasis.

IL-1 is a known inductor of the expression of adhesion molecules in cells. At a skin level, IL-1 stimulates the production of ICAM-1 and E-Selectin on endothelial cells [15] and the vasculature of the skin [16]. The expression of these molecules leads to the adherence of circulating leukocytes, favoring their transendothelial migration from the blood to the epidermis, perpetuating the inflammatory response in psoriasis [17,18]. Soluble adhesion molecules are present in the serum, GCF, and other body fluids [14,19]. They may be formed by alternative splicing mRNA and/or proteolytic cleavage [20], and their detection may reflect the state of endothelial activation in some diseases [21]. In this regard, higher levels of sICAM-1 and sE-selectin have been previously reported in the serum of psoriatic patients compared to healthy subjects. sICAM-1 levels have been reported to positively correlate with IL-18 psoriasis severity; this suggests that IL-18 could be regulating the production of sICAM-1 in psoriasis patients. [14,22]. Therefore, the detection of sICAM 1 and sE-selectin in bodily fluids could be used as prospective biomarkers of psoriasis.

At present, the diagnosis of psoriasis in the clinical setting is primarily based on the assessment of history, clinical features, and subjective measurements of patient’s satisfaction and quality of life. Treatment, on the other hand, is individualized depending on the severity of the disease. The lack of well-defined and standardized tools for severity assessment, combined with several clinical phenotypes of the disease and multiple differential diagnoses (including inflammatory, infectious, and neoplastic processes that mimic papulosquamous lesions), are major hurdles in the early diagnosis and timely treatment of psoriasis [23,24]. Skin biopsy and histopathology can be valuable for differential diagnosis; however, in addition to being an invasive procedure, it provides low sensitivity in the early stages of the disease [25]. Therefore, a supportive method for non-invasive diagnosis and treatment outcome evaluation is needed [26]. In this context, the detection of proteins within the gingival crevicular fluid (GCF) of psoriatic patients could be a useful resource for non-invasive diagnosis, treatment outcome evaluation, and the exploration of the immunoinflammatory pathogenesis of the disease.

Under physiological conditions, the GCF is a plasma transudate with a diverse composition naturally secreted at the gingival sulcus. Under pathological conditions, the GCF changes its composition, transforming into an inflammatory exudate. The GCF is used as a biological source of biomarkers to study periodontal and systemic diseases [27,28]. Recently, we reported downregulation of MMP-8 and overexpression of S100A8 in the GCF of patients with atopic dermatitis and psoriasis, respectively, independent of the patient’s periodontal status, indicating that MMP-8 and S100A8 proteins could be used as diagnostic biomarkers of these diseases, respectively [29,30]. Therefore, we hypothesize that psoriasis can modify the concentration of inflammatory biomarkers in the GCF. Consequently, this study aimed to compare GCF levels of IL-18, sICAM-1, and sE-selectin psoriatic subjects versus healthy controls, regardless of their periodontal status.

## 2. Materials and Methods

### 2.1. Study Design

The Scientific and Bioethics Committee of the Faculty of Dentistry of the Andrés Bello University, Santiago, Chile, approved the following research (no. PROPRGFO_002021_28). All subjects provided written informed consent before enrollment, in accordance with the ethical standards of the institutional and national research committees and the Helsinki Declaration. The STROBE (Strengthening the Reporting of Observational Studies in Epidemiology) guidelines for the reporting of cross-sectional studies were followed in the preparation of the final manuscript.

### 2.2. Participants

Volunteering subjects referred for presumed psoriasis (P) to the Dermatology Department of the San José Hospital, Santiago, Chile, were enrolled between March and December of 2018. At the same time, systemically healthy (HC) patients attending the Dental Clinic of the Faculty of Dentistry of Andrés Bello University, Santiago, Chile, were recruited as controls. The eligibility criteria for inclusion were (i) adults (>18 years old), (ii) with at least twelve teeth (excluding third molars), (iii) diagnosed with or without psoriasis. The exclusion criteria were (i) subjects with systemic and/or dermatological disorders aside from psoriasis (especially those involving immunoinflammatory dysregulation such as arthritis, systemic lupus erythematosus, and diabetes, to name a few); (ii) individuals who had received anti-inflammatory, antibiotic, and/or immunomodulatory treatment in the last three months; (iii) patients who had undergone radio- and/or chemotherapy in the previous year; (iv) pregnant females; and (v) patients who had received dermatological treatment for psoriasis and/or periodontitis in the last six months.

### 2.3. Patient Evaluations

All patients were examined by the same team of experts, as previously described [29]. Medical history and sociodemographic characteristics such as age, sex, and active smoking habit were recorded in predefined charts by dermatologists from the San José hospital staff. Patients with a previous history and/or under the suspicion of having undiagnosed non-communicable diseases were excluded from this research and immediately referred to a physician for further evaluation. Clinical diagnosis of psoriasis was based upon the presence of characteristic cutaneous lesions and histopathological evaluations. Psoriasis severity was defined using the (a) Psoriasis Area and Severity Index (PASI), (b) Body Surface Area (BSA), (c) Physician’s Global Assessment (PGA), and (d) Dermatology Life Quality Index (DLQI).

Periodontal status was defined following the joint clinical case definition proposed by the American Academy of Periodontology (AAP) and the Centers for Disease Control and Prevention (CDC) of the United States [31]. Clinical evaluations were performed by a calibrated periodontist from the faculty staff of Universidad Andrés Bello and included full-mouth periodontal charts and recording of the (a) probing depths (PD), (b) clinical attachment loss (CAL), and (c) the bleeding on probing index (BOP) at six sites per tooth excluding third molars (UNC-15 periodontal probe, Hu-Friedy, USA). Additional information regarding oral hygiene habits was recorded. Severe cases of periodontitis were defined as ≥2 interproximal sites with CAL ≥ 6 mm (not on the same tooth) and ≥ 1 interproximal site with PD ≥ 5 mm. Moderate cases of the disease were defined as ≥2 interproximal sites with CAL ≥ 4 mm (not on the same tooth) and/or ≥2 interproximal sites with PD ≥ 5 mm. Mild cases of periodontitis were defined as ≥2 interproximal sites with CAL ≥3 mm and ≥2 interproximal sites with PD ≥ 4mm (not on the same tooth) or one site with PD ≥5 mm. No periodontitis was defined as mild, moderate, or severe periodontitis. Finally, patients with a clinical diagnosis of periodontal disease were referred for treatment at the Periodontal Clinic of the San José Hospital or the Dental Clinic of the Faculty of Dentistry of Andrés Bello University.

### 2.4. Gingival Crevicular Fluid (GCF) Sampling

GCF samples were collected at the deepest site per quadrant as previously described [29]. Selected sites were carefully isolated with cotton rolls and gently air-dried with an air syringe to prevent saliva contamination. Sterile periodontal strips (Periopaper^®^, Oraflow, Plainview, NY, USA) were sub-gingivally placed into the gingival sulcus or pocket until mild resistance was noticed and left in place for 30 s. Subsequently, strips were collected into sterile tubes (Eppendorf^®^, Eppendorf AG, Hamburg, Germany) and immediately transferred for storage (−20 °C) and posterior analysis at the Periodontal Biology Laboratory of the Faculty of Dentistry of the University of Chile, Santiago, Chile.

### 2.5. Gingival Crevicular Fluid (GCF) Determinations

Pooled samples from each patient were prepared by adding forty microliters of protein buffer per strip into each sterile tube. Then, the elution was incubated for 30 min at 4 °C and centrifuged at 12,000 g for five minutes at 4 °C. The procedure was repeated twice, and samples were stored at −20 °C until further analysis. GCF aliquots from all subjects were used for protein quantification using a multiplex bead-based immunoassay (Human Magnetic Luminex Assay^®^, R&D Systems, Minneapolis, MN, USA) for IL-18, ICAM-1, and E-selectin following the manufacturer instructions. Finally, data from the multiplex panel were read using a dedicated platform (Magpix^®^, Millipore, MO, USA) and software (Milliplex AnalystR^®^, Viagene Tech, Minneapolis, MN, USA) according to the manufacturer.

### 2.6. Sample Size Calculation

Previously reported serum concentrations of IL-18 in patients with psoriasis and systemically healthy subjects were used to calculate the sample size requirements for this research [32]. Authors reported IL-18 levels of 43.7 ± 6.23 pg/mL and 32.2 ± 6.98 pg/mL in cases and controls, respectively. The determining effect size was 1.73 [32]. We estimated a more conservative effect size of 0.8, with a significance level of α = 0.05 and a power of 0.8. The results showed that the study needed a minimum sample size of 26 individuals per group. Post-hoc power analyses related to IL-18 and sE-selectin were performed, obtaining values of 0.996 and 0.998, respectively.

### 2.7. Statistical Analysis

Data analyses were performed using STATA v13^®^ StataCorp software (StataCorp. LLC, College Station, TX, USA), with a significance level of α = 0.05. Data distribution and homoscedasticity were examined using Levene’s test and the Shapiro–Wilk test. Inferential analyses were performed with Student’s t-test and Fisher exact test.

IL-18, sICAM-1, and sE-selectin concentrations were transformed to loge, fulfilling normality and homoscedasticity assumptions. The analysis was performed using a multiple linear regression model adjusted by periodontitis, age, and tobacco use. The performance discrimination and diagnostic accuracy for IL-18 and sE-selectin-1 were evaluated by constructing receiver operating characteristic (ROC) curves and estimating the area under the curve (AUC) of psoriasis patients versus controls. The threshold was calculated as the point of highest sensitivity and specificity in the ROC curve.

## 3. Results

In total, 81 subjects were included in this study: 42 patients with psoriasis and 39 systemically healthy controls. General dermatological and intraoral parameters are presented in Table 1. Patients in the case group presented moderate to severe psoriasis. No significant intergroup differences regarding gender, tobacco use, and BOP index were noticed (*p* > 0.05). The psoriasis group presented a higher mean age (*p* < 0.05) and worse periodontal parameters than controls (evidenced by higher PD, CAL, and an increased frequency and severity of periodontitis, *p* < 0.05).

GCF levels of IL-18 were significantly higher in the psoriasis group versus controls (*p* < 0.05). sE-selectin, on the other hand, was significantly lower in psoriasis patients compared to healthy subjects (*p* < 0.05). No intergroup differences in the levels of sICAM-1 were noticed (*p* > 0.05, Figure 1). Mean levels and standard deviations for IL-18 were 26.51 ± 10.46 pg/mL in psoriasis group and 18.65 ± 5.17 pg/mL in healthy control group; for sICAM-1, they were 28,647.45 ± 20,424.33 pg/mL in the psoriasis group and 24,334.59 ± 14,506.75 pg/mL in the healthy control group; for E-selectin, they were 31,490.35 ± 97,355.66 pg/mL in the psoriasis group and 201,873.5 ± 161,580.8 pg/mL in the healthy control group.

We further explored whether the presence of psoriasis influenced the GCF levels of IL-18, sE-selectin, and sICAM-1 after adjusting for age, tobacco use, and periodontitis severity (Table 2). According to our results, the diagnosis of psoriasis demonstrated a significant influence on the GCF levels of IL-18 and sE-selectin (*p* < 0.05), whereas mild periodontitis of the explored variables affected the concentrations of sICAM-1 (*p* = 0.02, results not shown).

Finally, the diagnostic precision of GCF IL-18 and sE-selectin for psoriasis was tested (Table 3). The receiver operating characteristic (ROC) curve for both biomarkers is shown in Figure 2. The ROC area for IL-18 was 0.77 (95%, C.I: 0.66–0.85), with a sensitivity and specificity of 73.81% and 64.10% for the diagnosis of psoriasis, respectively. sE-selectin, on the other hand, presented an ROC area of 0.68 (95%, C.I: 0.57–0.78), with a sensitivity and specificity of 90.48% and 61.54%, respectively.

## 4. Discussion

Psoriasis is a chronic immunoinflammatory dermatosis characterized by the presence of hyperactive keratinocytes secondary to Th1/Th17 immune dysregulation and persistent angiogenesis. These phenomena culminate with the formation of painful plaque/scaly lesions in visible skin areas, which in turn significantly impact the quality of life of those suffering the disease. In addition, severe forms of psoriasis have been associated with the development of low-grade systemic inflammation, an increased risk of cardiovascular disease, atherosclerosis, and other immunoinflammatory systemic diseases [33,34]. Even though psoriasis is a common disease with varied clinical manifestations and extensive research, there is a lack of non-invasive biomarkers available for early diagnosis and treatment outcome evaluation. Due to its local and systemic nature, it is crucial to assist clinicians with identifying, staging, and post-treatment evolution of psoriasis with, ideally, easy-to-perform and minimally invasive methods. According to our knowledge, this is the first study to demonstrate higher levels of IL-18 and lower levels of sE-selectin in the GCF in psoriatic patients compared to systemically healthy controls, thereby suggesting these proteins could be promising biomarkers for future evaluations and participate in the pathogenesis of psoriasis.

IL-18 is a potent proinflammatory cytokine produced and secreted by keratinocytes [35]. Previous studies suggest early participation of the cytokine in the inflammatory process and pathogenesis of psoriasis [36]. While strong epidermal hyperplasia and prominent Th1 inflammatory responses have been reported in psoriasis-like skin models [37], few studies have explored the in vivo concentrations of IL18 in psoriatic patients. In this study, we found higher concentrations of IL-18 in the GCF of psoriatic patients compared to controls. Our results are in line with previous publications reporting higher serum levels of IL-18 in psoriatic patients compared to systemically healthy controls [13,32,38]. In addition, a positive correlation between the serum concentrations of IL-18 and psoriasis severity and activity has previously been reported, hinting at the prospective value of cytokine as an objective biomarker for the disease [13,32]. At a local level, studies that explored the expression of IL-18 at psoriatic skin lesions have reported significantly higher interleukin levels in cutaneous lesions compared to healthy skin samples from the same dermatosis patients. Its expression was also reduced after dermatological treatment, reinforcing the participation of IL-18 in psoriasis pathogenesis [12].

In contrast, the exact role of IL-18 in periodontal tissues remains unclear. Animal models in rhesus monkeys have shown a downregulation of IL-18 mRNA in early periodontitis, suggesting that lower expression of the cytokine could be responsible for an impaired Th2 inflammatory response in periodontal disease [39]. In this sense, higher GCF concentrations of IL-18 have been reported in >6 mm periodontal pockets compared to periodontally healthy sites [40,41]; however, other studies failed to show significant differences in the cytokine GCF concentrations between periodontitis, gingivitis, and healthy subjects [42]. Nonetheless, in this context, it has been theorized that higher GCF concentrations of IL-18 could sustain periodontal inflammation and destruction, since IL-18 induces the in vitro formation of osteoclasts [43] and stimulates bone resorption and metalloproteinase secretion in human periodontal ligament fibroblasts (hPDLF) [44]. In addition, IL-18 has a role in nitric oxide synthesis and cell adhesion molecule expression, as the chemokine induces macrophage caspase 8 secretion and natural-killer cell IFNγ production [45]. Overall, we suggest that higher levels of IL-18 in the GCF in psoriatic patients could be derived from their serum as a reflection of the systemic inflammatory changes of the disease. However, more studies are still necessary to determine the role of IL-18 in periodontal tissue in psoriasis patients.

Transendothelial migration in inflammatory dermatoses such as psoriasis depends on the expression of E-selectin and ICAM-1 adhesion molecules on hematopoietic and endothelial cells [46,47]. Previous studies reported higher expression of both molecules in keratinocytes and endothelial cells from psoriatic plaques and healthy skin samples from psoriatic patients compared to systemically healthy controls [48]. E-selectin on the other hand was only expressed by endothelial cells, with higher concentrations in psoriatic lesions compared to non-involved psoriasis skin and systemically healthy controls [49]. Altogether, these findings suggest that ICAM-1 and E-selectin participate in the development of the different stages of the psoriatic plaque and sustain the inflammatory process of the skin.

Previous studies have reported higher serum levels of sICAM-1 and sE-selectin in psoriatic patients versus healthy controls [22,50,51]. In addition, both soluble molecules have been positively correlated with psoriasis activity [14,50], suggesting that the proteins could have a valuable potential as diagnostic and severity staging biomarkers for psoriasis. Our research group found lower GCF sE-selectin levels in psoriatic patients versus healthy controls, whereas no significant intergroup differences were noticed regarding the GCF concentrations of sICAM-1. To date, few studies have explored the GCF levels of sICAM-1 and sE-selectin in periodontal patients. A prior study found that active smokers and non-smoking periodontitis patients presented similar concentrations of GCF sICAM-1; nonetheless, serum concentrations of the soluble molecule were higher in the smoker group versus the non-smoker controls [52]. In another study, researchers reported similar GCF levels of total sICAM-1 and sE-selectin in the periodontal sulcus (PS ≤ 3 mm) of periodontitis patients and periodontally healthy controls after adjusting by age; however, higher amounts of both adhesion molecules were reported in the GCF of periodontal pockets in periodontitis patients compared to healthy sulcus from periodontally healthy controls [53]. In addition, no significant differences were found in the gingival capillary blood levels of sE-selectin between periodontitis and periodontally healthy subjects [54]. In a recent study, higher total amounts of GCF ICAM-1 were reported in subjects with Stage III-Grade C generalized periodontitis compared to healthy controls [19]. Although the exact role of soluble ICAM-1 and E-selectin molecules in periodontal tissues is still unclear, the GCF concentrations of the latter could be considered a local biomarker of inflammation and tissue destruction. Prior studies have reported that the serum concentrations of sICAM-1 and sE-selectin are higher in psoriatic patients compared to healthy controls; nonetheless, our study failed to find significant intergroup differences when sICAM was measured in the GCF of comparable patients. We believe this might be explained by the binding of these soluble molecules to their ligands on circulating cells in the serum or connective tissues, hence diminishing their transendothelial migration from the blood to the GCF of psoriatic patients. In fact, it has been reported that sICAM-1 usually binds to plasma fibrinogen and CD11a/CD18 receptors expressed on the vascular endothelium [55,56]. Therefore, sICAM-1and sE-selectin could participate in the development and immunomodulation of psoriatic skin lesions. However, more studies are needed to further understand the role of sE-selectin in the periodontal tissues of psoriatic patients.

Considering that the GCF protein content is influenced by periodontitis and tobacco consumption, a multiple linear regression model was performed to control for these potential confounding elements. According to our results, approximately 23% of IL-18 and E-selectin GCF levels could be explained by psoriasis in our models. A similar study demonstrated that the presence of apoptosis-associated speck-like proteins containing caspase-recruitment domains (which activate the effector caspase) could explain an estimated 25% of the increased serum concentrations of IL-18 observed in psoriatic patients. However, the GCF content could be modified by the presence of confounders, such as unaccounted comorbidities in psoriatic patients [38].

The clinical diagnosis of psoriasis is often challenging for inexperienced physicians, as skin lesions present in a subtle and nonspecific manner that resembles other dermatoses. To date, biomarkers for the disease are scarce; hence, the need to explore new biomarker sources has been widely acknowledged. In this context, our study explored the potential use of GCF as a novel, non-invasive, and safe source for new diagnostic biomarkers for psoriasis. Our results regarding the ROC curve area of IL-18 (0.77) in GCF samples were similar to those reported in previously in the serum samples [38]. To our knowledge, there are no ROC curve areas reported for the serum, saliva, and GCF levels of sE-selectin in psoriasis diagnosis. However, higher levels of serum sE-selectin in patients with oral squamous cell carcinoma (OSCC) and/or leukoplakia have been reported when compared to healthy controls. In that study, sE-selectin showed a 0.646 and 0.704 area under the ROC curve for the detection leukoplakia and OSCC, respectively [57]. Detecting biomarkers in the GCF has the advantage of being a less invasive, safe, and rapid method [13,30]. Thus, IL-18 and sE-selectin appear to be promising biomarkers for screening and diagnosis of psoriasis, helping clinicians make their early diagnosis. Considering that the most psoriasis biomarkers previously evaluated are universal for inflammation and psoriasis may be triggered also by genetic, epigenetic, infectious factors, among others [26], an imbalance of several biomarkers could be evaluated to improve the diagnostic accuracy. For this, we suggest continuing with the search for biomarkers in oral fluids comparing psoriasis severity at different stages, since they could reflect molecular changes in the progression of the disease, guiding the clinician to choose a personalized treatment. Additionally, it is important consider that psoriasis and atopic dermatitis are the most common skin diseases, so it would be relevant to compare the use of combined biomarkers in these two diseases to improve the accuracy of the diagnosis and evaluate their cost in relation to the use of a single biomarker.

This study presents the classic limitation of observational studies in the sense that it does not allow causality. It must be considered that despite our results, we cannot preclude that the concentrations of IL-18 and sE-selectin in the GCF of psoriatic patients may be modified to some extent by the presence of periodontitis, as the disease usually modifies the local composition of the GCF. In addition, we demonstrated that concentrations of IL-18 and sE-selectin in the GCF in moderate/severe psoriasis patients were upregulated and downregulated, with high standard deviations, respectively. The latter can be attributed to the different periodontal conditions of the subjects included. We suggest that the severity of the psoriatic patients recruited is capable of producing modifications in the serum concentrations of these adhesion molecules. In turn, plasma concentrations of the molecules also influence the normal protein content in the GCF. Finally, further studies are necessary to define the role of GCF proinflammatory cytokines and adhesion molecules in the pathogenesis of psoriasis and their utility as a diagnostic biomarker by clinicians.

## 5. Conclusions

Overall, the GCF levels of IL-18 and E-selectin were associated with moderate to severe psoriasis based on clinical and histopathological diagnoses. ICAM-1 levels, on the other hand, showed no significant intergroup differences between psoriasis and control groups. Considering their diagnostic precision in this research, we propose that the GCF measurement of these proteins could be used as valuable biomarker candidates for psoriasis diagnosis and severity staging of the disease in the future.

## Figures and Tables

**Figure 1 life-11-01000-f001:**
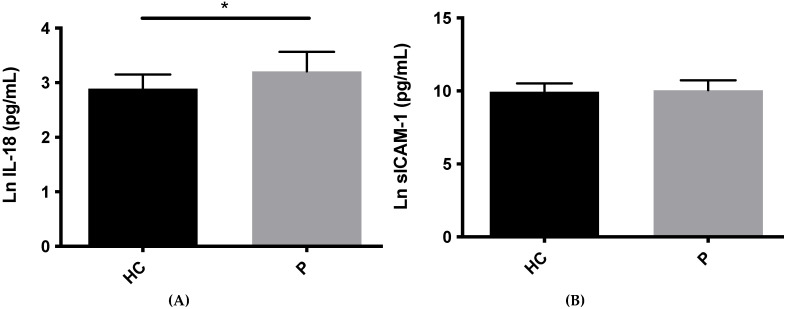
IL-18, sICAM-1, and sE-selectin levels (log natural scale) in the GCF in psoriasis patients and healthy controls. Ln: log natural scale; HC: healthy controls; P: psoriasis. (**A**) higher levels of IL-18 were observed in the GCF of patients with psoriasis compared to healthy controls; (**B**) No differences were found in the levels of sICAM-1 in the GCF between psoriasis and healthy controls; (**C**) lower levels of sE-selectin were showed in the GCF of patients with psoriasis compared to healthy controls. * *p* < 0.05.

**Figure 2 life-11-01000-f002:**
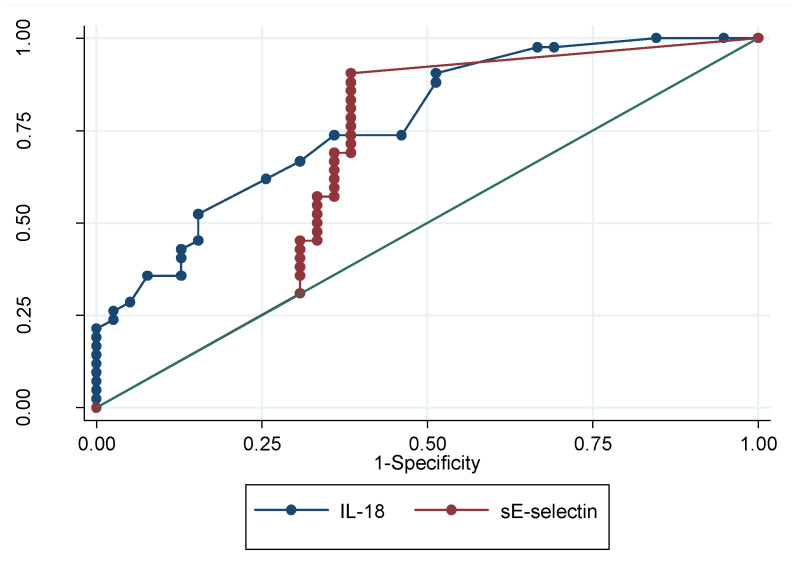
Receiver operating characteristic curves (ROC) for the GCF levels of IL-18 and sE-selectin in studied patients.

**Table 1 life-11-01000-t001:** General, dermatological, and intraoral parameters of psoriasis patients (P) and systemically healthy controls (HC).

Parameters	HC (n = 39)	P (n = 42)	*p*
Age (years, mean ± SD)	34.63 ± 12.03	47.5 ± 12.81	0.001
Gender: Female, male (%-n)	57.89–22, 42.11–16	42.86–18, 57.14–24	0.263
Smokers (%-n)	26.32–10	45.24–19	0.104
PD (mean ± SD)	1.89 ± 0.35	2.43 ± 0.50	0.001
CAL (mean ± SD)	1.39 ± 0.37	2.46 ± 1.28	0.001
BOP (mean ± SD)	12.36 ± 18.35	11.33 ± 13.15	0.779
Periodontitis: No periodontitis (%) Mild (%) Moderate (%) Severe (%)	48.72 12.82 33.33 5.13	7.14 14.29 40.48 38.10	0.001
PASI (mean ± SD)	-	10.90 ± 6.82	
BSA (%)	-	16.61 ± 15.01	
PGA (mean ± SD)	-	2.72 ± 0.73	
DLQI (%)	-	14.34 ± 7.64	

PD, probing depth; CAL, clinical attachment level; BOP, bleeding on probing index; PASI Psoriasis Area Severity Index; BSA, Body Surface Area Index; PGA, Physician’s Global Assessment; DLQI, Dermatology Life Quality Index, SD, standard deviation.

**Table 2 life-11-01000-t002:** Multiple regression models for IL-18 and sE-selectin levels in the GCF of psoriasis patients.

	IL-18 Levels		sE-Selectin Levels	
**Variables**	Coef. ± SE	*p*	Coef. ± SE	*p*
**Diagnosis**	0.21 ± 0.08	0.02	−3.29 ± 0.97	0.00
**Mild periodontitis**	−0.07 ± 0.12	0.57	−0.99 ± 1.29	0.44
**Moderate periodontitis**	0.01 ± 0.10	0.91	−1.80 ± 1.15	0.12
**Severe periodontitis**	0.14± 0.14	0.42	−1.64 ± 1.51	0.29
**Age (years)**	0.00 ± 0.00	0.93	0.03 ± 0.36	0.34
**Smoker**	0.11 ± 0.07	0.15	0.89 ± 0.80	0.27
**PD**	0.09 ± 0.09	0.30	−0.70 ± 0.94	0.46
**Constant**	2.66 ± 0.20		10.44 ± 2.14	
**Prob > F**	0.00		0.00	
**R-squared**	0.30		0.30	
**Adj. R^2^**	0.23		0.23	

Coef., coefficient; SE, standard error; GCF gingival crevicular fluid; PD, probing depth.

**Table 3 life-11-01000-t003:** Diagnostic precision of GCF levels of IL-18 and E-selectin.

Marker	Threshold	ROC Area	Sensitivity	Specificity	ROC Area (95% CI)
IL-18	19.57	0.77	73.81	64.10	0.66–0.85
E-selectin	994.4	0.68	90.48	61.54	0.57–0.78

## Data Availability

Not applicable.

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
