# Peer review of "Identification of IL-18 and Soluble Cell Adhesion Molecules in the Gingival Crevicular Fluid as Novel Biomarkers of Psoriasis"

_life, 2021, doi:10.3390/life11101000_

Round 1

Reviewer 1 Report

In this manuscript, Valenzuela et al. demonstrate differences in the levels of IL-18 and E-selectin in the gingival crevicular fluid in healthy patients and psoriatic patients, justifying its potential use as a non-invasive biomarker for psoriasis. This relationship held even across age, smoking status, periodontal health etc.

This report is interesting and potentially quite useful. Just a few comments for improvement.

The introduction is slightly lengthy and convoluted, but the authors do a fairly good job at walking the reader through why they chose to do the analyses contained in this manuscript.

Small issues of readability. The authors should go back over the manuscript carefully to fix minor awkward language usage.

“sICAM-1 levels have been reported to positively correlate with IL-18 psoriasis severity.” What do the authors mean by this?  

Authors y axes in figure 1 are labelled “natural log.” Authors should include units e.g. ln[IL-18] (pg/mL). Authors should also spell out in the caption what the groups stand for (HC is healthy control, P is psoriasis).

In the text, authors say “mean levels and standard deviations for IL-18, sICAM-1, and E-selectin in the psoriasis group were…   On the other hand…. In the healthy group were…”. It would be better and easier to compare numbers directly here if the authors instead listed the levels of IL-18 in psoriasis and healthy controls, and then listed the sICAM-1 for both groups, and then listed the sE-selectin levels.  

Author Response

Dear Reviewer,

Following your kind suggestion, we added the following to our manuscript:

“In this manuscript, Valenzuela et al. demonstrate differences in the levels of IL-18 and E-selectin in the gingival crevicular fluid in healthy patients and psoriatic patients, justifying its potential use as a non-invasive biomarker for psoriasis. This relationship held even across age, smoking status, periodontal health etc.”

This report is interesting and potentially quite useful. Just a few comments for improvement.

  1. The introduction is slightly lengthy and convoluted, but the authors do a fairly good job at walking the reader through why they chose to do the analyses contained in this manuscript.

A: Thank you for pointing this out. We agree with this comment: The introduction was summarized and some points explained.

  1. Small issues of readability. The authors should go back over the manuscript carefully to fix minor awkward language usage.

A: Agree. Manuscript was evaluated by English Editing Services from MDPI services.

  1. “sICAM-1 levels have been reported to positively correlate with IL-18 psoriasis severity.” What do the authors mean by this?  

A: To explain this result, we added: this suggests that IL-18 could be regulating the production of sICAM-1 in psoriasis patients.

  1. Authors y axes in figure 1 are labelled “natural log.” Authors should include units e.g. ln[IL-18] (pg/mL). Authors should also spell out in the caption what the groups stand for (HC is healthy control, P is psoriasis).

A: Thank you for this suggestion: The unit to the graphs and the explanation of the abbreviations were included.

  1. In the text, authors say “mean levels and standard deviations for IL-18, sICAM-1, and E-selectin in the psoriasis group were…   On the other hand…. In the healthy group were…”. It would be better and easier to compare numbers directly here if the authors instead listed the levels of IL-18 in psoriasis and healthy controls, and then listed the sICAM-1 for both groups, and then listed the sE-selectin levels.  

A: We agree with this and have incorporated your suggestion in the manuscript: The wording was changed as suggested: Mean levels and standard deviations for IL-18 were 26.51 ± 10.46 pg/ml in psoriasis group and 18.65 ± 5.17 pg/ml in healthy control group; for sICAM-1, they were 28647.45 ± 20424.33 pg/ml in psoriasis group and 24334.59 ± 14506.75 pg/ml in healthy control group; for E-selectin, they were 31490.35 ± 97355.66 pg/ml in psoriasis group and 201873.5 ± 161580.8 pg/ml in healthy control group.

Reviewer 2 Report

The study represents an interesting effort to identify biomarkers within novel fluids to be exploited for diagnostic purposes. On this subject, the study revealed interesting insights.

Concerning the sample size, although authors provided an a priori power analysis, they should also perform a post-hoc power analysis on the statistically significant results related to IL-18 and sE-selectin , which could enforce the results.

The Discussion section could be improved:

-How do authors explain the high standard deviations reported in lines 217-218?

-The analysis of a panel of several soluble markers and cytokines could improve the diagnosis. Do authors think that the analysis of more than three factors could be performed in a time and cost-effective manner?

-Moreover, could the integration of these biochemical factors with other molecular biomarkers (for instance, genetic and epigenetic factors) improve the diagnosis considering the multifactorial etiology related to Psoriasis?

-Finally, authors could discuss the importance of analyzing the biomarkers in different skin diseases (such as atopic dermatitis) and in more samples with different clinical severity (i.e. patients with mild, moderate, severe Psoriasis) in order to better evaluate their diagnostic power related to Psoriasis.

Author Response

Dear Reviewer,

We appreciate your new recommendation.

“The study represents an interesting effort to identify biomarkers within novel fluids to be exploited for diagnostic purposes. On this subject, the study revealed interesting insights”.

  1. Concerning the sample size, although authors provided an a priori power analysis, they should also perform a post-hoc power analysis on the statistically significant results related to IL-18 and sE-selectin, which could enforce the results.

A: You have raised an important point here. We added: We estimated more conservative effect size of 0.8, with a significance level of α=0.05 and a power of 0.8. The results showed that the study needed a minimum sample size of 26 individuals per group.  Post-hoc power analyses related to IL-18 and sE-selectin were performed, obtaining a value of 0.996 and 0.998, respectively.    

The Discussion section could be improved:

  1. How do authors explain the high standard deviations reported in lines 217-218?

A: Thank you for pointing this out. We added: In addition, we demonstrated that concentrations of IL-18 and sE-selectin in the GCF in moderate/severe psoriasis patients were upregulated and downregulated, with high standard deviations, respectively. The latter can be attributed to the different periodontal conditions of the subjects included.

  1. The analysis of a panel of several soluble markers and cytokines could improve the diagnosis. Do authors think that the analysis of more than three factors could be performed in a time and cost-effective manner?

A: Thank you for this suggestion, we answered considering point 5: For this, we suggest continuing with the search for biomarkers in oral fluids comparing psoriasis severity at different stages, since they could reflect molecular changes in the progression of the disease, guiding the clinician to choose a personalized treatment. Al-so, it is important consider that psoriasis and atopic dermatitis are the most common skin diseases, so it would be relevant to compare the use of combined biomarkers in these two diseases to improve the accuracy of the diagnosis and evaluate their cost in relation to the use of a single biomarker.

  1. Moreover, could the integration of these biochemical factors with other molecular biomarkers (for instance, genetic and epigenetic factors) improve the diagnosis considering the multifactorial etiology related to Psoriasis?

A: We added: Considering that the most psoriasis biomarkers previously evaluated are universal for inflammation and psoriasis may be triggered also by genetic, epigenetic, infectious factors, among others, an imbalanced of several biomarkers could be evaluated to im-prove the diagnostic accuracy.

  1. Finally, authors could discuss the importance of analyzing the biomarkers in different skin diseases (such as atopic dermatitis) and in more samples with different clinical severity (i.e. patients with mild, moderate, severe Psoriasis) in order to better evaluate their diagnostic power related to Psoriasis.
  • We added the information in point 3.

Round 2

Reviewer 2 Report

The revised version of the manuscript has improved.